# De-Speckling Breast Cancer Ultrasound Images Using a Rotationally Invariant Block Matching Based Non-Local Means (RIBM-NLM) Method

**DOI:** 10.3390/diagnostics12040862

**Published:** 2022-03-30

**Authors:** Gelan Ayana, Kokeb Dese, Hakkins Raj, Janarthanan Krishnamoorthy, Timothy Kwa

**Affiliations:** 1Department of Medical IT Convergence Engineering, Kumoh National Institute of Technology, Gumi 39253, Korea; gelan@kumoh.ac.kr; 2School of Biomedical Engineering, Jimma University, Jimma 378, Ethiopia; kokebdese86@gmail.com (K.D.); hakkinsbme@gmail.com (H.R.); 3Singapore Bioimaging Consortium, Astar Research Institutes, Singapore 138667, Singapore; 4Department of Biomedical Engineering, University of California, 451 Health Sciences, Davis, CA 95616, USA; 5Medtronic MiniMed, 18000 Devonshire St., Northridge, Los Angeles, CA 91325, USA

**Keywords:** ultrasound, filtering, speckle, clustering, block matching, non-local means

## Abstract

The ultrasonic technique is an indispensable imaging modality for diagnosis of breast cancer in young women due to its ability in efficiently capturing the tissue properties, and decreasing nega-tive recognition rate thereby avoiding non-essential biopsies. Despite the advantages, ultrasound images are affected by speckle noise, generating fine-false structures that decrease the contrast of the images and diminish the actual boundaries of tissues on ultrasound image. Moreover, speckle noise negatively impacts the subsequent stages in image processing pipeline, such as edge detec-tion, segmentation, feature extraction, and classification. Previous studies have formulated vari-ous speckle reduction methods in ultrasound images; however, these methods suffer from being unable to retain finer edge details and require more processing time. In this study, we propose a breast ultrasound de-speckling method based on rotational invariant block matching non-local means (RIBM-NLM) filtering. The effectiveness of our method has been demonstrated by com-paring our results with three established de-speckling techniques, the switching bilateral filter (SBF), the non-local means filter (NLMF), and the optimized non-local means filter (ONLMF) on 250 images from public dataset and 6 images from private dataset. Evaluation metrics, including Self-Similarity Index Measure (SSIM), Peak Signal to Noise Ratio (PSNR), and Mean Square Error (MSE) were utilized to measure performance. With the proposed method, we were able to record average SSIM of 0.8915, PSNR of 65.97, MSE of 0.014, RMSE of 0.119, and computational speed of 82 seconds at noise variance of 20dB using the public dataset, all with *p*-value of less than 0.001 compared against NLMF, ONLMF, and SBF. Similarly, the proposed method achieved av-erage SSIM of 0.83, PSNR of 66.26, MSE of 0.015, RMSE of 0.124, and computational speed of 83 seconds at noise variance of 20dB using the private dataset, all with *p*-value of less than 0.001 compared against NLMF, ONLMF, and SBF.

## 1. Introduction

Currently, the ultrasonic technique is a popular imaging modality for the diagnosis of breast cancer in young women with dense breast tissue [1,2,3]. However, ultrasound images are susceptible to noises, notably speckle noise. Speckle is a random noise pattern created by a large number of scattering waves from the tissues possessing random phases [4]. The scattering waves interfere in two ways: either detrimentally by creating speckles and mottled B-scan noises or constructively by creating intense noise [5,6]. Speckle noise in ultrasound images is characterized by its high frequency and poor visual or perception quality. It produces artificial structures while diminishing the actual boundaries of the tissue in breast ultrasound images. Furthermore, it poses challenges for the subsequent steps of the image processing pipeline, such as edge detection, segmentation, feature extraction, and classification [4,5]. Moreover, speckle is multiplicative in nature compared to other noises, which are usually additive; this makes it challenging to filter from ultrasound images [7,8].

Speckle reduction algorithms can be categorized into five types: (i) the local adaptive filters such as the Lee filter [9,10], the Frost filter [11,12] and the Bilateral filter [13,14]; (ii) the anisotropic diffusion filters such as the Detail Preserving Anisotropic Diffusion (DPAD) [15] and Oriented Speckle Reducing Anisotropic Diffusion (OSRAD) filters [16]; (iii) the multi-scale filters such as the Generalized Likelihood Method (GLM) filter [17], the Wavelet-Based Filter, and the Linear Wavelet filter [18]; (iv) the non-local means filters [19,20,21] such as the Optimized Bayesian Non-local Means (OBNLM) filter and the Probabilistic Patch-based (PPB) filter [22]; and (v) the hybrid filters such as the Non-local Means and Multi-scale Hybrid Filter [23,24], e.g., the SAR-Oriented Version of the Block Matching 3D (SAR-BM3D) filter.

Generally, the local adaptive filters have two disadvantages: being very sensitive to noise and enhancing the high contrast areas of images too much, resulting in the introduction of artifacts [25]. Anisotropic diffusion filters suffer from the need for intensive parameter adjustment (number of iterations), the tendency to degrade the fine structures in an image, and a reduction in the image resolution [26]. Multiscale methods are computationally intensive and require more constraints compared to other filters [17]. Similarly, non-local means filters suffer from an increased computational speed due to weighted averaging [21]. Finally, hybrid methods deteriorate the visual quality and do not preserve the edges in the images [24].

Therefore, despite the fact that the existing filtering methods reduce speckle noise significantly, there are still drawbacks associated with the filters in terms of achieving high fidelity breast cancer detection [25,26,27]. Firstly, these filters remove finer edge details that are necessary during diagnosis for defining the precise boundaries of tumors along with the speckles. During subsequent iterations, the filters also cause blurred contrast edges in the regions of low intensity, whereas the speckles are still retained in the high intensity regions [18]. Secondly, in most of the algorithms, the restored value of a pixel relies on the neighborhood pixels in its spatial vicinity; this is referred to as the locally adaptive recovery paradigm [26]. In contrast, the non-local methods do not entirely depend on neighborhood pixels; however, they require a longer processing or computation time compared to the local methods [20]. Finally, prior to de-noising, most of the algorithms do not distinguish pixel properties such as noise, speckle, or edge; hence, they cannot find an adequate balance between edge enhancement and the retention of small structures, especially when the quality of the source image is poor [18,26].

To overcome the shortcomings of the filter effects on speckle reduction in ultrasound images, a Multi-Layer Fusion Enhancement Method was proposed based on Block Matching and 3D (MLFE-BM3D) filtering [28]. This approach maintains a good filtering effect in the smooth region of the image and also improves the performance of the block matching as quantified by the Peak Signal-to-noise ratio (PSNR). Despite the improved noise reduction achieved by the MLFE-BM3D filter, drawbacks persist, such as the removal of finer edge details and the degradation of the edges and small dimensional structures, especially when the quality of the image is poor. Moreover, the processing time is relatively long and requires optimization.

Here we propose a de-speckling technique that uses rotationally invariant based block matching and non-local means (RIBM-NLM) filtering for de-speckling ultrasound breast cancer images for improved early diagnosis. The RIBM-NLM method includes pre-classification as well as the definition of new similarity terms; these techniques allow us to calculate a more reliable set of candidates required for calculating the weight factors used by the non-local means filtering based on block matching. To achieve the goal of finding more reliable sets of candidates, our method employs both pre-classification and the definition of a new similarity term. We want to increase the chance of finding candidates for non-repetitive patterns. Thus, pre-classification is used to provide candidate sets that can be from any part of the image. Moreover, because the moment invariants we use in pre-classification are rotationally invariant, the neighborhoods will potentially contain rotationally unaligned candidates. It is therefore necessary to define a new similarity term that can estimate the rotation angle during the matching process. This is where RIBM comes into play. Finally, the use of K-means increases the computation speed; this reduces the processing time.

## 2. Materials and Methods

### 2.1. The Denoising Process

The filtering pipeline starts with pre-processing via a Gaussian filter, followed by pre-classification using K-means clustering based on Hu’s moment invariants. Next, non-local means (NLM) filtering based on the rotationally invariant block matching (RIBM) is carried out as depicted in Figure 1. The Gaussian filter smoothens the image while making further operations scale invariant, and the clustering step improves NLM by identifying suitable candidates for the weighted averaging task [29]. The K-means clustering algorithm employed in the proposed method uses Hu’s moment invariants to group similar candidates within a cluster [30]. The RIBM process provides reliable, noise-tolerant, and rotationally invariant weights calculated for each cluster [31].

In the pre-classification process, the noisy input image is subjected to a Gaussian filter. Gaussian was chosen over other filters because the original pixel value receives the heaviest weight (having the highest Gaussian value), and the neighbouring pixels receive proportionally lower weights as their distance to the original pixel increases. This results in a blur that preserves boundaries and edges better than other more uniform blurring filters. Furthermore, utilizing the Gaussian filter allows visual operations to be made scale invariant, which is necessary for dealing with the size variations that may occur in image data. This is because the images may be of different sizes and, in addition, the distance between the object and the acquisition method may be unknown and may vary depending on the circumstances. In general, the important properties of the Gaussian blur that made it appropriate in our case include Gaussian kernel linearity, shift invariance, semi-group structure, non-enhancement of local extrema, scale invariance, and rotational invariance.

Consider a (2m+1)×(2m+1) square mask, with a center (0,0) and x,y ranges from (−m,−m) to (m,m). The element of the Gaussian mask is given by Equation (1):(1)$$G_{\sigma }\left(x,y\right)=e^{\left(\frac{-\left(x^{2}+y^{2}\right)}{2\sigma^{2}}\right)}$$
where σ is the standard deviation of the Gaussian distribution. To keep the brightness level balanced in the image, we have performed normalization using $Sum_{\sigma }$, Equation (2), as in Equation (3):(2)$$Sum_{\sigma }=\sum_{x=-m}^{m}\sum_{y=-m}^{m}G_{\sigma }\left(x,y\right)$$
(3)$$G_{k\sigma }\left(x,y\right)=\frac{G_{\sigma }\left(x,y\right)}{Sum_{\sigma }}$$
where $G_{k\sigma }$ is the normalized Gaussian filter used for generating the Gaussian blurred ($G_{b}$) output image as in Equation (4):(4)$$G_{b}=G_{k\sigma }*\nu$$
where ν denotes the intensity of the input noisy image and ∗ denotes the convolution operation.

In this work, we do not use a larger value of σ because it introduces additional artifacts, and we want to retain most of the details of the input noisy image. After the Gaussian filter is applied to the input noisy image, the resulting blurred image is used as an input for the following clustering-based pre-classification process (see Appendix A).

The Gaussian filtered image is then converted into patches and used in the subsequent processes. For each of the patches, Hu’s moment invariant features are calculated. In our analysis, Hu’s moment invariants of order 2 have been used, and a feature descriptor of dimension: (1×7), a row vector was calculated for each patch. These feature descriptors are used as input for K-means clustering as in [31]. Consider N×N image and n×n patch with center i (i=1,2,…,N×N). The moment invariants and feature metrics ($\phi_{1},\phi_{2},\cdots ,\phi_{7}$) are calculated for each patch and represented as a 1×7 row vector. Then, for the entire image, there exist N×N feature vectors. These vectors are given as the input to the K-means clustering and the N×N vectors are clustered into K groups using the objective function in Equation (5):(5)$$\begin{array}{c} argmin\\ c \end{array}\overset{K}{\underset{k=1}{\sum }}\sum_{\begin{array}{c} H\left(Gb\left(i\right)\right)\epsilon Hm_{k}\\ i=1,2,\ldots ,N\times N \end{array}}{\left|H\left(Gb\left(i\right)\right)-\mu_{k}\right|}^{2}$$
where Gb(i) is a Gaussian blurred image patch with center i. The H(·) provides the moment invariants feature vector for the input patch, whereas $\mu_{k}$ is the mean vector for the kth cluster, $Hm_{k}$. Consequently, we obtain K clusters,$\;\left\{Hm_{1},\;Hm_{2},Hm_{3},\ldots ,Hm_{k}\right\}$, with each cluster $Hm_{i}$ containing $l_{i}$ feature vectors. Therefore, each cluster has a different length, $l_{i}$. In general, a pre-classified feature vector is represented with indices k and l as $Hm_{kl}$, where the indices span: k=1,…,K;l=1,…,L. Here, the index k corresponds to different clusters, and the index l corresponds to different patches that are clustered within it [32].

Most image processing methods use various parameters to recognize the different features present in an image. The objective here is to find several such numerical parameters that better characterize the image features. Statistical moments are usually used to generate moment invariants that describe structural or shape features. The moment invariants are normalized so that the intensity differences in images do not affect performance. These shape descriptors are used by the K-means clustering algorithm to cluster similar patches. Furthermore, tables corresponding to irregularity, compactness, aspect ratio, and size are generated from these moment invariants. The K-means clustering algorithm clusters N patches of an image into K classes based on the feature descriptors provided to it. In our case, the objective of clustering is to reconstruct a given patch using a small number of the other patches least corrupted by noises. Consequently, we achieve this by classifying an image into N small patches and looking for a close match for each patch within the pre-classified K image templates; we can then send a closer and better fit for the image in the form of a list of matching templates with the labels k1, k2,…, kN. Clustering-based pre-classification performs faster without the loss of any pixels in the weight calculation.

K-means clustering was chosen over fuzzy C-means clustering and other clustering methods due to two advantages: it processes quickly without eliminating any pixels during the weight calculation and clusters data into only a single cluster, with no overlapping clusters. Fuzzy C-means clustering, in contrast to K-means clustering, clusters data into multiple clusters. In our case, we want each pixel of the image to be clustered only into a single group, not into multiple groups. This property of K-means clustering helps to find unique pixels for the weighted averaging process that follows, thereby increasing the speed and quality of speckle reduction. In the K-means algorithm, the patches are partitioned into distinct clusters, and every member of a patch is possessed by exactly one cluster [33]. This prevents the candidates from being available in more than one cluster. Here, K-means clustering supplies the preselected candidates for the upcoming weighted averaging. The classification data, which are the coordinates of the block centre, are kept in the look-up table (LUT). Afterwards, the weighted averaging is carried out within each cluster (Appendix A). In summary, following the pre-classification, the original noisy image is clustered into K classes, and a look-up table is generated. Afterwards, the weighting average is carried out.

The rotationally invariant block matching process uses the LUT and calculates the weighted averaging for NLM filtering [31]. Non-local means filtering is based on the fact that images have patches that possess self-similarity. Consider a noisy image, v, with intensity, v(i), at each pixel, i, or coordinates (x,y) as follows in Equation (6):(6)v={v(i)|i∈Ω,Ω⊂ℝ2}

The NLM intensity at each pixel, i, represented as NL(v)(i), is nothing but the weighted average of $i^{\prime }s$ neighborhood pixels, I, denoted as Equation (7):(7)$$NL\left(v\right)\left(i\right)=\sum_{j\in I}w\left(i,j\right)v\left(j\right)$$
where v represents the input Gaussian blurred image, v(j) is the intensity at each pixel j, and w(i,j) is the weightage factor calculated for each v(j) in order to restore the intensity of the noise corrupted intensity v(i) at pixel i. The weightage factor for pixel i is calculated from all its neighborhood pixels j∈I. The weightage factor w(i,j) is calculated as in Equation (8):(8)$$w\left(i,\;j\right)=\frac{1}{Z\left(i\right)}\left(e^{-\frac{\|\nu \left(N_{i}\right)-\nu {\left(N_{j})\right\|}_{2,\alpha }^{2}}{h^{2}}}\right)$$
where $N_{i}$ and $N_{j}$ denote two different patches of fixed size whose central pixels are i, j, respectively. $\|\nu \left(N_{i}\right)-\nu {\left(N_{j})\right\|}_{2,\alpha }^{2}$ is a similarity measure, which is the Gaussian weighted Euclidean distance between the pixels i and j. The factor α>0 defines the width of the Gaussian function used, which in turn determines the weightage factor applied to the Euclidean distance. The variable h is the width of a Gaussian filter function used in the calculation. The normalization constant, Z(i), used in Equation (8) is given by Equation (9):(9)$$Z\left(i\right)=\sum_{j\in I}w\left(i,j\right)$$

For the conventional NLM, the neighborhood pixels, j∈I, are defined as all the pixels present in the image. The patches for the pixels i and j are defined as a circle whose center pixels are i and j, respectively, with radius r. The similarity term ${\left\|.\right\|}_{2}^{2}$ or the Euclidean distance is calculated between each pixel of the two patches being compared, weighted with respect to the Gaussian filter of the width factor α and summed up. The normalization factor, Z(i), is the sum of all the weightage factors, calculated for a single pixel, i, with respect to all pixels, j∈I.

In the proposed method, instead of calculating the weightage factor $w_{R}\left(i,\;j\right)$ for all pixels, j∈I, the patches with high self-similarity are pre-classified into K groups using K-means clustering. After this classification, for an N×N image and an n×n patch size with a center (i=1,2,…,N×N), the number of patches in each cluster is given by the set $l=\left\{l_{1},l_{2},\cdots ,l_{k}\right\}$. The proposed NLM, ($NLM_{p}$), is given by Equation (10):(10)$$NLM_{p}\left(v\right)\left(i\right)=\sum_{j\in L}w_{R}\left(i,j\right)v\left(j\right)$$

Here, the weightage factor, $w_{R}\left(i,j\right)$, is calculated by comparing the patch i with every other patch that was clustered together with the patch i, which is represented as j∈L.

The modified weight $w_{R}\left(i,\;j\right)$ is given by Equation (11):(11)$$w_{R}\left(i,\;j\right)=\frac{1}{Z_{R}\left(i\right)}e^{-\frac{d_{R}\left(i,j\right)}{h^{2}}}$$

Here, $w_{R}\left(i,\;j\right)$ is defined through a distance metric, $d_{R}\left(i,j\right)$, which is explained in (28). This way, the computational time is minimized by carrying out the calculation of weights within each cluster rather than the entire image.

Grewenig et al. applied moment invariants to enhance block matching that only improved the matching capability of NLM within a spatial neighbourhood centred around the target patch. In our case, the matching candidates for a given patch are defined from a set of different patches that originate from the entire image. In order to improve the weight calculation in the NLM filtering based on block matching, the rotation angle between two patches has to be estimated from its moment invariants (Appendix A). The invariant moment of order, p,q, required to evaluate the feature vectors is given by Equation (12):(12)$$\mu_{p,q}=\sum_{y\in F}\sum_{x\in F}{\left(x-x_{C}\right)}^{p}{\left(y-y_{C}\right)}^{q}$$
where x,y∈F denotes the x and y coordinates of all pixels of a feature F. Moreover, p,q∈N are different powers for the two dimensions, and $x_{c}$ and $y_{c}$ are the centroid coordinates where $\mu_{1,0}$ = $\mu_{0,1}$
*=*
$\mu_{1,1}$ = 0. Furthermore, $\mu_{0,0}$ is the number of pixels, which is area of the shape. Hu et al. [30] have defined seven feature metrics, $\phi_{1}$ through $\phi_{7}$, that are rotation-invariant and are defined as Equations (13)–(19).
(13)$$\phi_{1}=\mu_{2,0}+\mu_{0,2}$$
(14)$$\phi_{2}={\left(\mu_{2,0}-\mu_{0,2}\right)}^{2}+4\mu_{1,1}^{2}$$
(15)$$\phi_{3}={\left(\mu_{3,0}-3\mu_{1,2}\right)}^{2}+{\left(3\mu_{2,1}-\mu_{0,3}\right)}^{2}$$
(16)$$\phi_{4}={\left(\mu_{3,0}+\mu_{1,2}\right)}^{2}+{\left(\mu_{2,1}+\mu_{0,3}\right)}^{2}$$
(17)$$\begin{array}{ll} \phi_{5}=\left(\mu_{3,0}-3\mu_{1,2}\right) & \left(\mu_{3,0}+\mu_{1,2}\right)\left[{\left(\mu_{3,0}+\mu_{1,2}\right)}^{2}-3{\left(\mu_{2,1}+\mu_{0,3}\right)}^{2}\right]\\ & +\left(\mu_{2,1}+\mu_{0,3}\right)\left(\mu_{0,3}+\mu_{2,1}\right)\left[3{\left(\mu_{1,2}+\mu_{3,0}\right)}^{2}-{\left(\mu_{2,1}+\mu_{0,3}\right)}^{2}\right] \end{array}$$
(18)$$\begin{array}{ll} \phi_{6}=\left(\mu_{2,0}-\mu_{0,2}\right) & \left[{\left(\mu_{1,2}+\mu_{3,0}\right)}^{2}-{\left(\mu_{2,1}+\mu_{0,3}\right)}^{2}\right]\\ & +4\;\mu_{1,1}\left(\mu_{3,0}+\mu_{1,2}\right)\left(\mu_{0,3}+\mu_{2,1}\right) \end{array}$$
(19)$$\begin{array}{ll} \phi_{7}=\left(3\mu_{2,1}-\mu_{0,3}\right) & \left(\mu_{1,2}+\mu_{3,0}\right)\left[{\left(\mu_{1,2}+\mu_{3,0}\right)}^{2}-3{\left(\mu_{2,1}+\mu_{0,3}\right)}^{2}\right]\\ & +\left(3\mu_{2,1}-\mu_{3,0}\right)\left(\mu_{2,1}+\mu_{0,3}\right)[3{\left(\mu_{1,2}+\mu_{3,0}\right)}^{2}-{\left(\mu_{2,1}+\mu_{0,3}\right)}^{2}] \end{array}$$

The seventh moment of Hu, $\phi_{7}$, is used to check whether two patches are mirrored images of each other because the sign of $\phi_{7}$ changes only under mirroring, and is invariant to rotation or the presence of noise.

The process of RIBM is summarized as follows. Given that patch $N_{j}$ is a mirrored form of $N_{i}$, (i.e., the signs of $\phi_{7}$ for $N_{j}$ and $N_{i}$ are not same), then we can mirror $N_{j}$ at an arbitrary axis to obtain $N_{j}^{\prime }$; else, $N_{j}$ is the same as $N_{j}^{\prime }$.
Estimate the rotation angle between $N_{i}$ and $N_{j}$;For each pixel in $N_{i}$, find the corresponding pixel in $N_{j}$ via rotation by the estimated angle;The sum of the intensity differences in pixels $N_{i}$ and corresponding pixels $N_{j}$ is the required distance.


Then, the patch centroid is calculated in Equation (20). We have $N_{j}$, which is a noisy as well as a rotated patch with respect to the patch $N_{i}$. For defining the centroid, we suppose that the pixels within a patch are controlled by a coordinate system that has its center at the patch’s center.
(20)$$c_{i}∶=\left(\begin{array}{c} \frac{\int_{i}x_{b}\cdot v(x_{b},\;y_{b})\;dx_{b}\;dy_{b}}{\int_{i}v(x_{b},\;y_{b})\;dx_{b}\;dy_{b}}\\ \frac{\int_{i}y_{b}\cdot v(x_{b},\;y_{b})\;dx_{b}\;dy_{b}}{\int_{i}v(x_{b},\;y_{b})\;dx_{b}\;dy_{b}} \end{array}\right)$$
where $v\left(x_{b},y_{b}\right)$ denotes the intensity value of the patch $N_{i}$, and $\vec{c_{i}}$ denotes the normalized vector of the centroid of $N_{i}$. In the numerator of the above equations, the intensity, $v\left(x_{b},y_{b}\right)$, is weighted by the values of its coordinates, $x_{b}$ or $y_{b}$, to obtain the centroid, $c_{i}=\left(\begin{array}{c} c_{x}\\ c_{y} \end{array}\right)$.

To carry out the rotation prior to block matching, we define the rotation matrix from the elements of the centroid, $c_{i}$ and $c_{j}$, of block i and j, respectively. Since mirroring has to be compensated, we define a vector, $m_{i,j}\left(u\right)$, expressed as in Equation (21):(21)$$m_{i,j}\left(u\right)∶=\{\begin{array}{cc} \left(\begin{array}{c} -u_{x}\\ u_{y} \end{array}\right), & \mathrm{if}\;\phi_{7}\left(i\right)\cdot \phi_{7}\left(j\right)<0\\ \left(\begin{array}{c} u_{x}\\ u_{y} \end{array}\right), & \mathrm{else} \end{array}$$

Suppose the patch i,j, exhibits mirroring; then, $m_{i,j}\left(u\right)$ changes the sign of $u_{x}$, i.e., the x component; else, the components of u are used as they are.

To rotate and align the candidate patches to the target patch, $N_{i}$, it is necessary to estimate the rotation angle between $N_{i}$ and each candidate, $N_{j}\in \left(N_{k,1},N_{k,2},\cdots ,N_{k,l-1},N_{k,l}\right)$, that is present within the same cluster, k. Every pixel of a block can be represented as a vector from the block’s center; thus, the entire block or its corresponding vectors should be rotated by an equal angle. Therefore, the rotation angle between two blocks’ centroids $\vec{c_{i}}$ and $\vec{c_{j}}$ has to be estimated to rotate the block itself. The rotation matrix required to rotate the candidate block, j, to align with the target block, i, is given by Equation (22):(22)$$R_{i,j}=R_{\vec{c_{i}}}^{-1}\cdot R_{m_{i,j}\left(\vec{c_{j}}\right)\;}$$

The rotation matrices on the right-hand side are defined as in Equation (23):(23)$$R_{u}∶=\left(\begin{array}{c} u_{1}-u_{2}\\ u_{2}\;u_{1} \end{array}\right)$$

Using Equations (20) and (23), we can explicitly write, $R_{\vec{c_{i}}}^{-1}$ as in Equation (24):(24)$$R_{\vec{c_{i}}}^{-1}∶={\left(\begin{array}{c} c_{x}-c_{y}\\ c_{y}\;c_{x} \end{array}\right)}^{-1}$$

And using Equations (21) and (23), we write $R_{m_{i,j}\left(\vec{c_{j}}\right)}$ as in Equations (25) and (26) for mirrored and non-mirrored cases:(25)$$R_{m_{i,j}\left(\vec{c_{j}}\right)\;}∶=\begin{array}{ccc} \left(\begin{array}{c} -c_{x}-c_{y}\\ c_{y}-c_{x} \end{array}\right), & \; & \mathrm{if}\;\mathrm{mirrored} \end{array}$$
(26)$$R_{m_{i,j}\left(\vec{c_{j}}\right)\;}∶=\begin{array}{ccc} \left(\begin{array}{c} c_{x}-c_{y}\\ c_{y}\;c_{x} \end{array}\right), & \; & \mathrm{if}\;\mathrm{non}-\mathrm{mirrored} \end{array}$$

Subsequently, the corresponding point, $q_{i}$, defined as any point in patch $N_{j}$, is rotated to a corresponding point, $q_{j}$, in another patch $N_{j}$, using the rotation matrix as in Equation (27):(27)$$q_{j}=m_{i,j}\left(R_{i,j}\cdot q_{i}\right)$$

The resultant vector that comes out of the product of $R_{i,j}\cdot q_{i}$ is again compensated for the mirroring effect using the mirror function, $m_{i,j}\left(.\right)$, in Equation (21).

Lastly, the similarity term that replaces the Euclidean metric $\|\nu N\left(i\right)-\nu N{\left(j)\right\|}_{2,\alpha }^{2}$ in the conventional NLM weights can be computed by Equation (28):(28)$$d_{R}\left(i,j\right)=d_{R}\left(i,j\right)=\sum_{q_{i}\in i}{\left(v_{i}\left(q_{i}\right)-I_{n}\left(v_{j},q_{j}\right)\right)}^{2}$$$I_{n}$ defines the bilinear interpolation. The interpolation is required because, after rotation of the $N_{j}$, the intensity at non-integral pixel values needs to be evaluated for patch comparison, so extrapolation of the intensities from the integer pixel values to non-integer pixel values is required.

### 2.2. Experimental Setup and Materials

For the purpose of evaluating the proposed method against state-of-the-art filters for speckle noise reduction, the switching bilateral filter (SBF) [34], the non-local-means filter (NLMF) [20], and an optimized non-local means filter (ONLMF) [27] were employed on publicly available ultrasound images and private images.

The image dataset used for this study was obtained from the publicly available Mendeley dataset composed of 250 breast ultrasound images, of which 150 are malignant cases, and 100 are benign cases [35]. This study also utilized a private dataset with 6 images of patients who had breast imaging and a biopsy carried out at Black Lion Hospital (BLH), Ethiopia. The ultrasound images were acquired using the SonoScape Ultrasound (S60 model, SonoScape Medical Corp, Shenzhen, Guangdong, China) diagnostic unit equipped with a 7.5 MHz transducer.

The RIBM-NLM method was implemented using MATLAB software (MATLAB 2017a, MathWorks, Natick, MA, USA), and the experiments were performed on a personal computer that runs on a 2.5 GHz Intel(R) Core (TM) i3 (HP 15-dw model, Hewlett-Packard company, Palo Alto, CA, USA) processor.

This study develops a clustering method based on moment invariants. The conventional K-means clustering requires three main parameters to be decided: the type of distance measurement used, the number of clusters to be assigned, and the size of the vectors to be used in the NLM based framework. For measuring the distance between two feature vectors, we implement the Euclidean based distance as in [29]. We define the patch size to be 15 × 15 as in [36]. The changing trends of PSNR and MSE are roughly the same: when K (the number of clusters) becomes larger, there are more clusters representing different types of details. However, if K is too high, some clusters will not have enough candidates, and that degrades the image to be reconstructed. As a result, the PSNR and MSE decrease after the climax. Therefore, if complexity is not a concern, we can choose the optimal value of K depending on the size of the input noisy image. With this hypothesis, we performed a preliminary experiment to choose the optimal K value for our method. When K increases, the rates of change of PSNR and MSE reach a maximum and start to decline (Figure 2). When the value of K is much larger, then clusters with insufficient number candidates result, which degrades the quality of the reconstructed image. This results in a poor score for PSNR, which starts to decrease after a maximum is reached, as shown in Figure 2. Therefore, we have to choose an optimal value for K, which is sub-optimal in terms of the PSNR value. In Figure 2, even though the K value corresponding to the maximal PSNR score is 800, we set K = 675, which is a sub-optimal value. This value of K is chosen by considering the image size used for the study, which is 225 × 225, and the number of candidates that are supposed to be in one cluster. Additionally, the computational time taken for processing the images when K = 800 is 1.8 times greater than when K = 675. Time is the important constraint in our study because one of our targets is to decrease the processing time.

In our experiment, we use a Gaussian blur with standard deviations σ of 10, 20, and 50. The smoothing parameter, h, is kept at 12σ to allow a fair comparison of all methods. The σ in Equation (1) is fixed to 0.5 × σ, and the radius of the block size is set to m = 4.

Three widely used quantitative metrics are applied to evaluate the performance of the proposed speckle reduction method against the others. These are structural similarity index (SSIM), peak signal-to-noise ratio (PSNR) and mean squared error (MSE).

The SSIM is defined as in Equation (29):(29)$$SSIM\left(x,y\right)=\frac{\left(2\mu_{x}\mu_{y}+C_{1}\right)\left(2\sigma_{xy}+C_{2}\right)}{\left(\mu_{x}^{2}+\mu_{y}^{2}+C_{1}\right)\left(\sigma_{x}^{2}+\sigma_{y}^{2}+C_{2}\right)}$$

Here, x and y are two non-negative images, i.e., the initial noisy image and the de-noised image, respectively. $\mu_{x}$ and $\mu_{y}$ are the mean intensities of the image x and y, respectively. $\sigma_{x}^{2}$ and $\sigma_{y}^{2}$ are the variances of the intensities of images x and y, respectively; $\sigma_{xy}$ is the co-variance computed from the intensities of images x and y. $C_{1}\;\mathrm{and}\;C_{2}$ are constants introduced to avoid the instability of dividing by zero in the denominator factors of Equation (29) when $\mu_{x}^{2}+\mu_{y}^{2}$ and $\sigma_{x}^{2}+\sigma_{y}^{2}$ are too close to zero. Values of SSIM range from zero to one, and a higher value indicates a better de-noising effect.

The PSNR is defined as in Equation (30):(30)$$PSNR=10\;\log_{10}\left(\frac{L_{D}^{2}}{MSE}\right)$$

Here, $L_{D}$ is the magnitude of the difference between the maximum and the minimum intensity value, and MSE is the mean squared error between the original and the reconstructed images. The PSNR is a measure of the signal-to-noise ratio variations within an image. A higher value of PSNR indicates a better de-noising performance.

The mean squared error (MSE) is defined as in Equation (31). The smaller the MSE, the better the quality of image is.
(31)$$MSE=\frac{\sum_{i=1}^{row}\sum_{j=1}^{column}{\left(x\left(i,j\right)-y\left(i,j\right)\right)}^{2}}{Row\times Column}$$

The root mean squared error (RMSE), which is the square root of the mean squared error, is calculated for each image.

Furthermore, processing time in seconds, t(s), is used to evaluate the computational speed of the proposed method relative to the other three methods.

Moreover, the statistical significance of the results obtained using the proposed method compared to the-state-of-the-art methods is evaluated for the above three metrics (SSIM, PSNR, and MSE) using the *t*-test *p*-value pair-wise comparison method [37].

## 3. Results

### 3.1. Private Dataset Image Results

Figure 3 shows the visual performance of our method compared to the switching bilateral filter (SBF), the NL-means filter (NLMF), and an optimized non-local means filter (ONLMF) at σ=20. It can be observed that the visual quality of the RIBM-NLM method (Figure 3b) is superior relative to the SBF (Figure 3e), retaining sharp boundaries with fewer artifacts. The NLMF (Figure 3d) and the ONLMF (Figure 3c) methods, on the other hand, show significant distortion in the image with a reduction in the speckle and background noises. Thus, the RIBM-NLM method shown in Figure 3b performs better in terms of preserving edges and speckle noise suppression compared to other methods.

From Table 1, it can be inferred that the RIBM-NLM method has the highest value of SSIM (0.8915, compared to 0.7594 for ONLMF, 0.7284 for NLMF, and 0.8271 for SBF), which demonstrates that the proposed method’s performance is better than the other methods in terms of speckle reduction. Therefore, the RIBM-NLM method generates images with better structural similarity compared to the other three methods.

Figure 4 shows (see Appendix A) that the RIBM-NLM method scores the highest PSNR value and the smallest MSE, as well as the fastest time, when compared to the other three methods. Particularly, the PSNR value of the RIBM-NLM method is 3 dB higher than that of SBF method for noisy images whose σ is less than 50 (Figure 4a). Furthermore, the MSE value of the proposed method is the smallest compared to the other three methods, with a *p*-value of less than 0.001 (Figure 4b). Similarly, the proposed method provided a better RMSE value than the other three methods (see Figure 4c). Regarding the computational run time, the NLMF, ONLMF, SBF, and RIBM-NLM methods consume an average of 179 s, 138 s, 130 s, and 82 s, respectively, at a noise factor of 20 (Figure 4d). The processing time for the RIBM-NLM is the fastest, with a *p*-value of less than 0.001 relative to other methods; this is the result of the K-means clustering implemented in the pre-classification stage.

### 3.2. Private Dataset Image Results

Evaluations were also carried out on private breast cancer images obtained via ultrasound. Figure 5a is the original image obtained from the private dataset. Figure 5b–e shows the results obtained by the RIBM-NLM, ONLMF, NLMF and SBF methods, respectively. It can be observed that the pixel intensity of the image filtered by our algorithm is smoother than those of the other three methods. The better performance of the RIBM-NLM method is evident from the improved edges and the effective smoothening of the speckle noise in the private ultrasound image.

Table 2 shows the superiority in performance of the RIBM-NLM method in terms of visual quality; our method has the highest SSIM value at 0.8307.

From Figure 6 (see Appendix A), it can be observed that the PSNR values of the RIBM-NLM filter for all the clinical images (blurred with different levels of noise factor) are higher than those of the other filters. This demonstrates the robustness of the RIBM-NLM filter method in the presence of different noise levels. The PSNR value for the RIBM-NLM filter is 3% higher than that of the other filters and retains maximum edge details in the images (Figure 6a). The MSE values for the RIBM-NLM method at different noise levels are smaller than that of NLMF, ONLMF, and SBF methods (Figure 6b). It has been observed that the RIBM-NLM filter produces an MSE value 6% less than that of the SBF filter with a minimal degradation in the image quality. The same is also true for the RMSE values (see Figure 6c). Additionally, the RIBM-NLM method has the shortest processing time compared to the other methods when processing clinical images, as shown in Figure 6d.

The PSNR, MSE, RMSE, and t(s) scores obtained for the different clinical images processed using the RIBM-NLM method are shown in Table 3. It can be easily shown that the proposed method provides consistent results in terms of PSNR, MSE, and time for all the images.

## 4. Discussion

In this study, a combination of clustering-based pre-classification and rotationally invariant block matching for non-local means filtering is proposed for speckle reduction in breast ultrasound images. In doing so, the focus is on developing a better de-speckling method that is simple in clinical applications with less computing complexity and a short execution time while preserving the details of the image. Our method achieves promising results, outperforming the state-of-the-art methods by providing better SSIM, PSNR, and MSE scores. This is mainly because our approach applies filters in both the spatial and frequency domains by including block matching on top of the non-local means filter. The clustering algorithm, on the other hand, enables us to identify proper candidates within a short period of time; additionally, the application of block matching preserves the details needed for further processing. Equipped with these advantages, the NLM filter performs better at reducing speckle. Our approach provides the NLM method with the right candidates to replace every pixel with the weighted average of other pixels with similar neighborhoods. The most time-consuming part of the NLM filter is the weight calculation, where many of the available methods try to identify and eliminate dissimilar patches before weighted averaging. RIBM-NLM overcomes this challenge by using clustering-based pre-classification that minimizes the time required for finding the candidates that are used in the weight calculation. The main difference between the NLM method and other spatial approaches is that the weights in the NLM filter do not depend on the spatial distance between the target patches and the candidates, but depend mainly on the difference in intensity values. Previous studies that applied moment invariants to enhance block matching only improved the matching capability of NLM within a spatial neighborhood that is centered on the target patch. In our case, the matching candidates for a given patch are defined from a set of different patches that originate from the entire image.

Speckle noise affects the quality of ultrasound images quite significantly and is difficult to remove. This study is of practical significance because it reduces speckle noise in ultrasound images, resulting in ultrasound images with less noise interference and improved quality, preserving the necessary structures and resolvable details.

In this study, the parameters of the non-local means filter were fixed and used as a non-adaptive filter while processing images. Additionally, analysis was performed only on a small number of private images due to a limitation in the availability of data. Therefore, future studies will include adaptive filters applied to large datasets to improve the performance of the analysis.

## 5. Conclusions

In this paper, we proposed a rotationally invariant block matching (RIBM-NLM) method for de-speckling breast ultrasound images. The reduction in speckles achieved using the proposed method on breast ultrasound images obtained from public and private databases was compared with other methods such as ONLMF, NLMF, and SBF using SSIM, PSNR, and MSE. Results show that the RIBM-NLM method effectively reduces speckle while retaining finer details, as indicated by the MSE value that is 6% less than the state-of-the-art methods. RIBM-NLM also shows superior performance in terms of de-speckling compared to the existing filters, as indicated by the PSNR value, which is 3 dB higher, especially for the images of poor quality with a large σ value. Finally, the computation time consumed by the RIBM-NLM method is small compared to the ONLMF, NLMF, and SBF methods, with a duration 1.6 times shorter than that of SBF. This work is of significant importance for breast cancer early diagnosis in low resource settings where ultrasound is used as a primary means of breast cancer diagnosis.

## Figures and Tables

**Figure 1 diagnostics-12-00862-f001:**
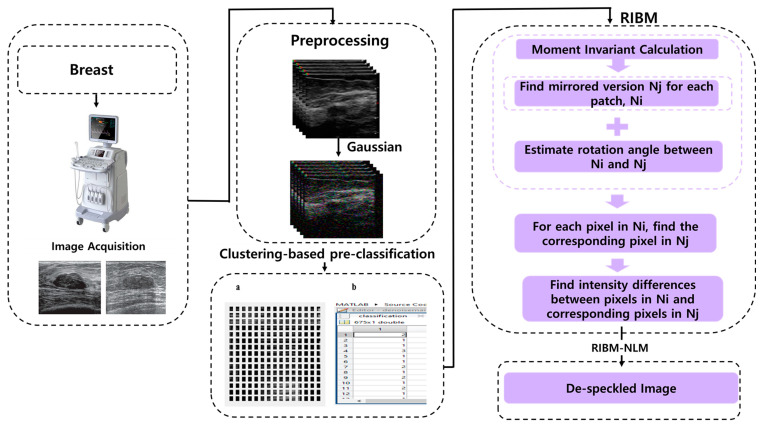
Processing pipeline of the RIBM-NLM method. (**a**) Patch separation; (**b**) K-means clustering. $N_{i}$,$\;N_{j}$ denote two different patches of fixed size whose central pixels are i, j, respectively.

**Figure 2 diagnostics-12-00862-f002:**
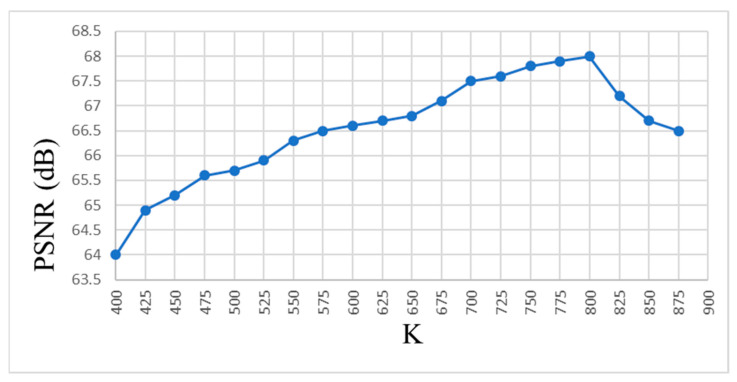
Peak signal-to-noise ratio (PSNR) vs. number of clusters K.

**Figure 3 diagnostics-12-00862-f003:**
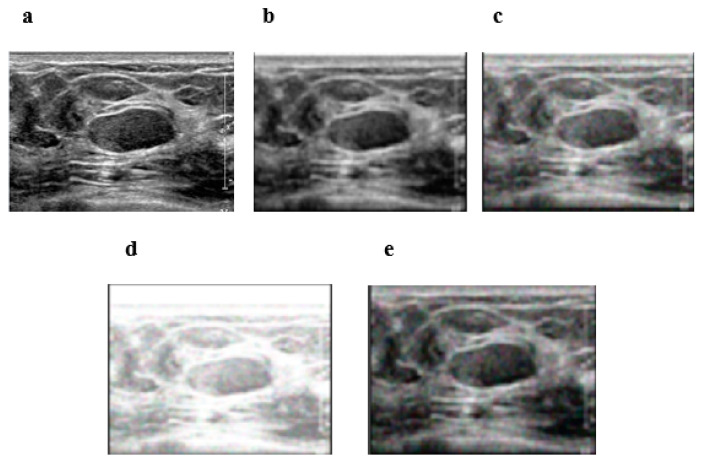
Visual results for database data. Performance of the four methods at σ = 20. (**a**) Original image; (**b**) RIBM-NLM method; (**c**) ONLMF; (**d**) NLMF; (**e**) SBF. RIBM-NLM, rotationally invariant block matching based non-local means filter; NLMF, non-local means filter; ONLMF, optimized non-local means filter; SBF, switching bilateral filter.

**Figure 4 diagnostics-12-00862-f004:**
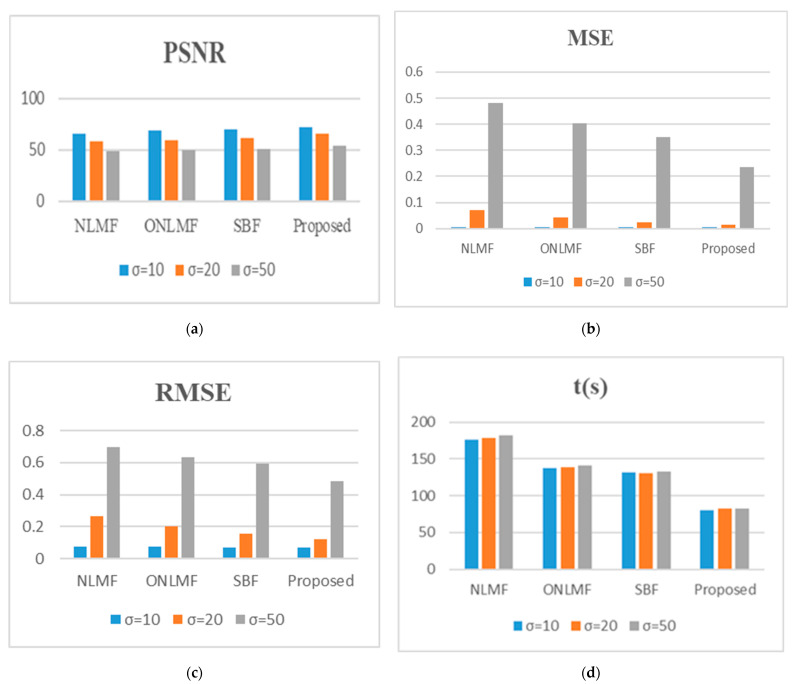
Average (**a**) PSNR; (**b**) MSE; (**c**) RMSE; (**d**) t(s) for several algorithms using public database images. NLMF, non-local means filter; ONLMF, optimized non-local means filter; SBF, switching bilateral filter; PSNR, peak signal-to-noise ratio; MSE, mean squared error; RMSE, root mean squared error; t(s), time in seconds; σ, Gaussian blur standard deviation.

**Figure 5 diagnostics-12-00862-f005:**
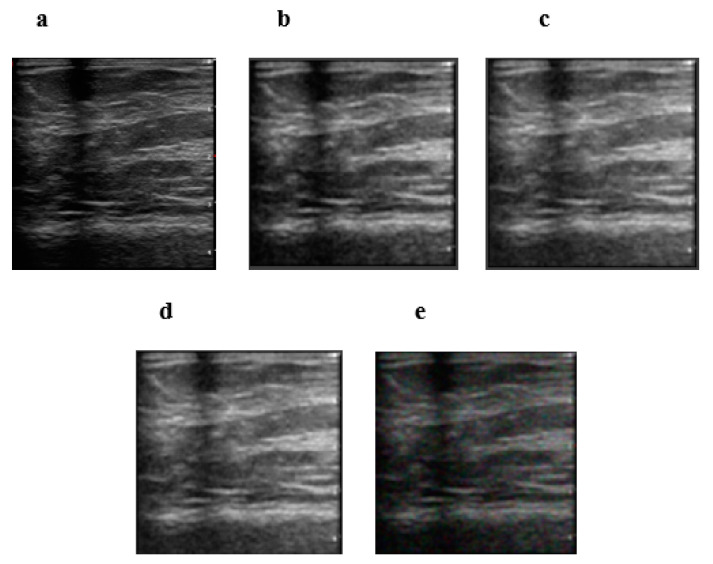
Visual results for clinical data. Performance of the four methods at σ = 20. (**a**) Original image; (**b**) RIBM-NLM method; (**c**) ONLMF; (**d**) NLMF; (**e**) SBF. RIBM-NLM, rotationally invariant block matching based non-local means filter; NLMF, non-local means filter; ONLMF, optimized non-local means filter; SBF, switching bilateral filter.

**Figure 6 diagnostics-12-00862-f006:**
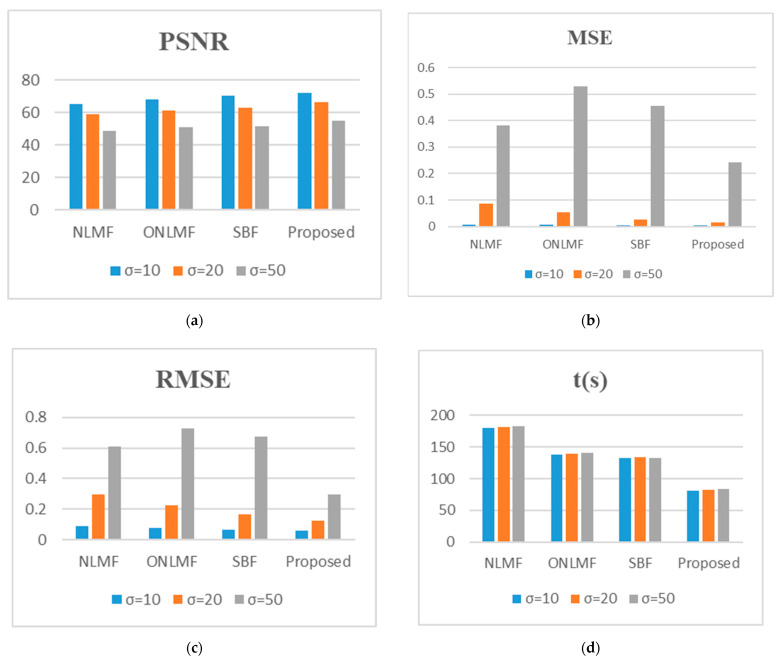
Average (**a**) PSNR; (**b**) MSE; (**c**) RMSE; (**d**) t(s) for several algorithms using private database images. NLMF, non-local means filter; ONLMF, optimized non-local means filter; SBF, switching bilateral filter; PSNR, peak signal-to-noise ratio; MSE, mean squared error; RMSE, root mean squared error; t(s), time in seconds; σ, Gaussian blur standard deviation.

**Table 1 diagnostics-12-00862-t001:** SSIM values for several algorithms using database images. NLMF, non-local means filter; ONLMF, optimized non-local means filter; SBF, switching bilateral filter; SSIM, self-similarity index metrics; σ, Gaussian blur standard deviation.

Metric	Proposed	ONLMF	NLMF	SBF
SSIM, at σ = 20	0.8915	0.7594	0.7284	0.8271

**Table 2 diagnostics-12-00862-t002:** SSIM values for several algorithms using clinical images. NLMF, non-local means filter; ONLMF, optimized non-local means filter; SBF, switching bilateral filter; SSIM, self-similarity index metrics; σ, Gaussian blur standard deviation.

Method	Proposed	ONLMF	NLMF	SBF
SSIM, at σ = 20	0.8307	0.7241	0.6963	0.8148

**Table 3 diagnostics-12-00862-t003:** Averaged quantitative de-noising results for six clinical images using the proposed method. CI1, clinical image 1; CI2, clinical image 2; CI3, clinical image 3; CI4, clinical image 4; CI5, clinical image 5; CI6, clinical image 6; Av., average; PSNR, peak signal-to-noise ratio; MSE, mean squared error; RMSE, root mean squared error; t(s), time in seconds; σ, Gaussian blur standard deviation.

Image	σ											
		σ=10				σ=20				σ=50		
	PSNR	MSE	RMSE	t(s)	PSNR	MSE	RMSE	t(s)	PSNR	MSE	RMSE	t(s)
CI1	71.3869	0.003314	0.057567	81.003529	66.9903	0.016732	0.129352	82.858238	55.146	0.282528	0.531533	83.405858
CI2	72.8369	0.00339	0.058223	80.160006	66.0016	0.011358	0.106573	82.474946	55.8126	0.269671	0.519298	84.678944
CI3	71.594	0.003678	0.060646	82.375192	65.2058	0.015101	0.122886	82.564148	55.0384	0.236662	0.486479	83.601501
CI4	72.7233	0.003046	0.055190	82.34266	66.9023	0.019146	0.138369	83.244738	54.2856	0.228034	0.477529	84.899632
CI5	72.177	0.003978	0.063071	80.703136	65.7659	0.016782	0.129545	82.174985	54.7928	0.20081	0.448118	81.204046
CI6	72.1992	0.003072	0.055425	82.167858	66.7179	0.014479	0.120328	83.648587	54.4104	0.237003	0.486829	84.764421
Av.	72.15288	0.003413	0.058354	81.4587301	66.263966	0.0155996	0.124509	82.827607	54.9143	0.2424513	0.491631	83.759067

## Data Availability

In this study, we used publicly available breast ultrasound images, Mendeley ultrasound dataset (https://data.mendeley.com/datasets/wmy84gzngw/1 (accessed on 15 March 2022)). The private images can be made available for reasonable requests by contacting the corresponding authors.

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
