# Peer review of "De-Speckling Breast Cancer Ultrasound Images Using a Rotationally Invariant Block Matching Based Non-Local Means (RIBM-NLM) Method"

_diagnostics, 2022, doi:10.3390/diagnostics12040862_

Round 1

Reviewer 1 Report

Overall, this is a very well written paper outlining a new method to improve on breast cancer ultrasound technique. This is a very important topic as breast ultrasound is one of the most common modalities to image the breast for breast cancer diagnosis.

I have very minor edits to recommend.

Abstract:

  1. Remove nowadays from the first sentence. It portrays a more relaxed feeling to an otherwise formal paper.
  2. Line 26 would read better if stated In this study, we propose a breast ultrasound...

Introduction

  1. Line 42: ...breast cancer in young women with dense breast tissue.

Methods

  1. Line 281: ...image dataset used for this study was obtained...

Results

  1. Line 374: Evaluations have also...
  2. Line 433: Remove the comma after that

Conclusions

1. Line 449: In this paper, we proposed a rotational...

Author Response

Response to Reviewer 2 Comments

Authors would like to thank the reviewer for the valuable inputs for the amelioration of the quality of our manuscript. Authors agree with the concerns raised by the reviewer and have made the necessary modifications as per the suggestions. Listed below are our responses to the questions and suggestions given by the reviewer 2. The reviewers’ comments are presented in “Black” while the responses are in “Red”. The updated parts in the manuscript are presented highlighted in “Yellow”.

Point 1: The key words should be different from those used in the title.

Response 1: We are grateful for the correction and the keywords have been updated. (See line 38)

ultrasound; filtering; speckle; clustering; block-matching; non-local means

Point 2: Please check the references cited in the text (e.g. lines 263, 269)

Response 2: Thank you for poiting this out. Those were supposed to be equation numbers and now we have corrected them in the manuscript. (See lines 268, 269, 283, 340)

Using (20) and (23), we can explicitly write,   as in (24).

And using (21) and (23), we write as in (25) and (26) for mirrored and non-mirrored cases.

The resultant vector that comes out of the product of  is again compensated for mirriong effect using the mirror function,  , (21).

,, are the variances of the intensities of images  and , respectively; whereas, , is the co-variance computed from the intensities of images  and .  are constants introduced to avoid instability of dividing by zero in the denominator factors of (29), when  and  are too close to zero.

Additional clarifications

In addition to the above comments, minor spell check has been done and corrections have been made.

Sincerely,

Authors

March 19, 2022

Reviewer 2 Report

General comment: The authors presented an interesting and original work, concerning to the de-speckling of breast cancer ultrasonography images.

The manuscript is written in a comprehensive way.

Title: The title is adequate.

Abstract: It is adequate.

The keywords should be different from those used in the title.

Introduction: It is adequate. The authors provided an adequate overview of the thematic.

Materials and methods: The methods are properly described.

Results: The Results are clearly described and supported by the Figures and Tables.

Discussion: It is adequate.

Conclusion: The conclusion is based on the results.

References: Please check the references cited in the text (e.g. lines 263, 269).

Author Response

(The authors gave the same response as above.)

Reviewer 3 Report

The abstract needs modification. Section 1 has to state the need for this method. Section 2 needs total revision. The references need to be improved with latest addition . From equation 13 to 19 are they representing approximate representation of the coefficients. why only K means why not Fuzzy c means? The inference from Figure 2 has to be enhanced and analyzed. Find out other parameters to evaluate the results other than PSNR, MSE and SSIM. The conclusion needs correction.

Round 2

Reviewer 3 Report

All the corrections are included in the manuscript by the authors.